# Diagnosis of Alzheimer's Disease Based on the Modified Tresnet

**Zelin Xu, Hongmin Deng \*, Jin Liu and Yang Yang**

School of Electronics and Information Engineering, Sichuan University, Chengdu 610065, China;
zelin_xu@stu.scu.edu.cn (Z.X.); liu_jin@scu.edu.cn (J.L.); argus@stu.scu.edu.cn (Y.Y.)
\* Correspondence: hm_deng@scu.edu.cn

**Abstract:** In the medical field, Alzheimer's disease (AD), as a neurodegenerative brain disease which is very difficult to diagnose, can cause cognitive impairment and memory decline. Many existing works include a variety of clinical neurological and psychological examinations, especially computer-aided diagnosis (CAD) methods based on electroencephalographic (EEG) recording or MRI images by using machine learning (ML) combined with different preprocessing steps such as hippocampus shape analysis, fusion of embedded features, and so on, where EEG dataset used for AD diagnosis is usually is large and complex, requiring extraction of a series of features like entropy features, spectral feature, etc., and it has seldom been applied in the AD detection based on deep learning (DL), while MRI images were suitable for both ML and DL. In terms of the structural MRI brain images, few differences could be found in brain atrophy among the three situations: AD, mild cognitive impairment (MCI), and Normal Control (NC). On the other hand, DL methods have been used to diagnose AD incorporating MRI images in recent years, but there have not yet been many selective models with very deep layers. In this article, the Gray Matter (GM) Magnetic Resonance Imaging (MRI) is automatically extracted, which could better distinguish among the three types of situations like AD, MCI, and NC, compared with Cerebro Spinal Fluid (CSF) and White Matter (WM). Firstly, FMRIB Software Library (FSL) software is utilized for batch processing to remove the skull, cerebellum and register the heterogeneous images, and the SPM + cat12 tool kits in MATLAB is used to segment MRI images for obtaining the standard GM MRI images. Next, the GM MRI images are trained by some new neural networks. The characteristics of the training process are as follows: (1) The Tresnet, as the network that achieves the best classification effect among several new networks in the experiment, is selected as the basic network. (2) A multi-receptive-field mechanism is integrated into the network, which is inspired by neurons that can dynamically adjust the receptive fields according to different stimuli. (3) The whole network is realized by adding multiple channels to the convolutional layer, and the size of the convolution kernel of each channel can be dynamically adjusted. (4) Transfer learning method is used to train the model for speeding up the learning and optimizing the learning efficiency. Finally, we achieve the accuracies of 86.9% for AD vs. NC, 63.2% for AD vs. MCI vs. NC respectively, which outperform the previous approaches. The results demonstrate the effectiveness of our approach.

**Keywords:** deep learning; Alzheimer's disease; multi-receptive-fields; structural MRI

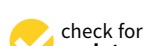

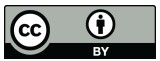

## 1. Introduction

In 1906, the German psychiatrist Alois Alzheimer introduced a result about the diagnosis of the August Deter disease, which was the embryonic form of Alzheimer's disease. After more than 100 years of development, Alzheimer's disease (AD) has been clearly defined [1,2].

AD is a very common neurodegenerative brain disease characterized by the degeneration of specific nerve cells, presence of neuritic plaques, and neurofibrillary tangles. It causes confusions in memory, cognitive function, and behavior, and the damages are

irreversible [3]. Alzheimer patients suffer from confusions, difficulties in adapting to their environments, problems related to speech and language skills, low motivation problems, and so on. The average survival time for AD patients is five years. It is important to diagnose Alzheimer's disease early [4]. According to Sohu report: China has become the country with the largest number of AD patients in the world. Due to the growth of aging society, AD will become a global burden. By 2050, it is expected to rise to 131.5 million patients. In other words, there will be one affected AD patient per 85 persons by 2050 [5,6], and the cost of treatment and care for AD patients is also expected to increase significantly. In the mean time, mild cognitive impairment (MCI), which is known as the prodromal stage of AD, is an intermediate process in the conversion of normal people to AD. There are up to 15 percent of people with MCI being converted to AD each year [7,8].

Diagnosis of Alzheimer's disease is a challenging task. Generally, the AD detection and progression are dealt with by a variety of clinical neurological and psychological tests, and is confirmed through electroencephalographic (EEG) and magnetic resonance imaging (MRI) [9]. However, recently, many methods using only EEG or MRI data have become a hot topic in the computer-aided (CAD) diagnosis of AD. The EEG-based AD detection mainly incorporates machine learning methods like support vector machines etc, while deep learning techniques have seldom been applied on large EEG datasets. Less MRI dataset was not only applied in ML methods [10], but also suitably applied in deep learning methods. In the magnetic resonance imaging-driven method, hippocampus shows the highest rate of atrophy during illness. Therefore, MRI has become a hot target for early diagnosis of AD. A method for automatically identifying Alzheimer's disease by MRI images is in urgent need of development.

In last decades, most detection methods of AD are dependent on clinical observations, in which AD can be diagnosed but not predicted until the condition gets worse. In recent years, computer-aided diagnosis (CAD) methods are increasingly used based on ML approaches clinicians to evaluate the patient data and diagnose the diseases cardia [11]. More recently, CAD based on deep learning (DL) has become a common technique for the early diagnosis of AD as a new research direction [12]. DL is utilized to simulate the hierarchical structure of human brain and process data from low level to high level, where data of high level is the inherent law and representation level of learning sample data [13].

The main contributions of this paper are listed in the following two aspects: (1) We proposed a novel framework based on the modified Tresnet neural network to diagnose AD. (2) The process of the imbalanced data is carried on by transfer learning and 5-fold cross-validation experiments.

The purpose of our study is to use only the MRI images to diagnose AD in the framework of deep neural network. And the proposed approach differs from the early studies in the following aspects: (1) We extract the Gray Matter images in MRI as the inputs of the network rather than the original MRI images. (2) The fusion technology of multi-receptive-field features is added to the deep network model. (3) Transfer learning and 5-fold-cross-validation experiments are added for handling impact caused by small and imbalance dataset.

The rest of the paper is organized as follows: In Section 2, we discuss the related work about AD classification. In Section 3, the procedure and the key technology of our approach are described. The evaluation metrics and experimental of our approach are shown in Section 4. Further discussion is carried out in Section 5, and conclusion is drawn in the last Section.

## 2. Related Work

Recently, an automated system that assists AD diagnosis has attracted more and more attention of researchers [14,15]. Machine learning (ML) is one of the most important methods, more and more scholars utilize ML to diagnose AD. For example, Zhang, Y.P. et al. proposed selection and fusion of embedded features strategy based on SVM, where

the model was based on the MRI data of T1 weighted images in AD vs. Normal Control (NC), and its accuracy reached 87.7% [16].

In 2007, Li et al. analyzed the shape of hippocampus based on ML. This dataset included 39 subjects split as 19 AD, 20 NC, it reached an accuracy of 85% [17]. It was found that AD would cause greater variations in the shape of the left hippocampus than that of the right hippocampus, so its effect of diagnosing AD was better.

Deep learning is an important branch of ML, many scholars have tried to input preprocessed medical data into various deep neural networks to train an auxiliary diagnosis model, and better results have been achieved.

In Gunawardena's research, a three dimensional (3D) convolutional neural network (CNN) model based classification method was proposed. The accuracy for classification of three categories among AD, NC, and MCI reached 96% [18].

In the first half of 2017, Aderghal et al. used two dimensional (2D) images of the hippocampus in different coordinate systems (Sagittal, Coronal, Axial), and employed three CNNs for feature extraction from MRI images in three coordinates systems, which was followed by a fully connected layer for classification. This dataset included 815 subjects split as 188 AD, 228 NC, and 399 MCI. The final accuracy for classification of AD vs. NC reached 85.8%, MCI vs. NC reached 65% and AD vs. MCI was 67% [19].

Korolev et al. proposed two different convolutional neural networks and compared them with each other by using the method of 3D subject-level. The first was the common network with convolutional layer and pooling layer, the second was the modern residual neural network (Resnet). In the experiment, there were 50 AD and 61 NC, and the accuracy reached 80% without data leakage (input $1 \times 110 \times 110 \times 110$) [20].

In the study of Valliani and Soni (2017), 188 AD, 243 MCI, and 229 NC subjects were collected to train the Resnet network. The Resnet model was pretrained on ImageNet data, then this model parameters were transfered to reduce the problem of little MRI data, and affine transformation was carried out on the data through rotation, flipping, and translation for data augmentation. The accuracy for binary classification of AD vs. NC reached 81.3%, and the more difficult 3-way classification (AD vs. MCI vs. CN) reached only 56.8% [21].

In Cheng's research, the innovation was that it further adopted the joint classification of Positron Emission Computed Tomography (PET) and MRI. First, PET and MRI medical graphics were sent to two 3D CNN networks, respectively, then the output features extracted from the two networks were inputted into an ordinary 2D CNN network. The accuracy of this muti-modality was greater than that of using PET or MRI images alone, reached 89% in AD vs. NC. The accuracies of the model reached 87.1% and 85.5% by using PET or MRI alone, respectively [22].

In 2018, Lin et al. proposed a method to analyze the correlation of each position of MRI image for the diagnosis of AD. A patch of size $40 \times 40 \times 40$ with value 0 was used to cover an area of the MRI Gray Matter (GM) in the study. They concluded that the distribution of relevance varied between patients, some patients pay more attention to the temporal lobes, whereas for others more cortical areas were involved. The dataset included 193 AD and 151 NC, and the accuracy reached 79% [23].

In 2019, Kanghan et al. used convolutional autoencoder based unsupervised learning for the AD vs. NC classifcation task, and supervised transfer learning was applied to diagnose AD. And a visualization method based on gradient was proposed. In the experiment, the dataset contained 198 AD and 230 NC, the classification result was 86.6%, 88.5%, 84.5% at accuracy (ACC), sensitivity (SEN), specificity (SPE) [24].

However, in these studies, the data in the training set and validation set might be from the same subject, and the testing set was not independent from the training set and validation set due to the partition errors in the dataset [25]. In other words, the same subject as in the training set might be also in the validation set, and the features of the testing set might have been trained, it was called data leakage. So it lacked feasibility despite the high precision [18]. In Wen's research (2020), they summarized several main causes of data leakage, such as wrong data split, late split, biased transfer learning, absence of

an independent test set. The dataset contained 336 AD and 330 NC. Then the 3D subject-level CNN, 3D ROI-based, 3D patch-level CNN, and 2D slice-level CNN were established, respectively. Under the situation of no data leakage, the accuracies of 3D subject-level CNN, 3D ROI-based, 3D patch-level CNN, 2D slice-level CNN, and support vector machine (SVM) were 83.5%, 87%, 78%, 82%, 88% separately [25].

The previous classification methods based on deep learning were generally realized by using the most basic CNN framework to adjust parameters or simply adding a few convolutional layers, pooling layers and fully connected layers. The range of model selection was very limited, and the model improvement was also simple and single, such as LeNet-5, VGGNet, Resnet, and so on. In this article, we diagnose AD based on modified Tresnet network. Tresnet improves on the following five aspects: SpaceToDepth stem, Anti-Alias downsampling (AA), In-Place Activated BatchNorm, Blocks selection, and SE module. The Tresnet had higher accuracy and efficiency than the previous ConvNets. The experimental steps are as follows: First FSL software and batch processing commands on the computer terminal are used to register the MRI images with heterogeneities, then the MRI images are segmented by MATLAB tool kits to obtain the Gray Matter images, next the Gray Matter images are inputted into the Tresnet network. Meanwhile, in order to increase the classification effect, the Squeeze-and-Excitation (SE) module in the Tresnet network is replaced by the Selective Kernel (SK) module, so that the convolutional layer can receive features from multiple receptive fields. With the slight increase in model complexity during optimization, the classification effect is better than the state of the art study. Finally, due to the small and imbalanced dataset, the 5-cross-validation and transfer learning methods are adopted to speed up and optimize the learning efficiency of the proposed model.

## 3. Materials and Methodology

The flowsheet of the proposed classification framework based on deep learning for distinguishing the patients among AD, MCI, and NC clinical statuses by MRI data is shown in Figure 1.

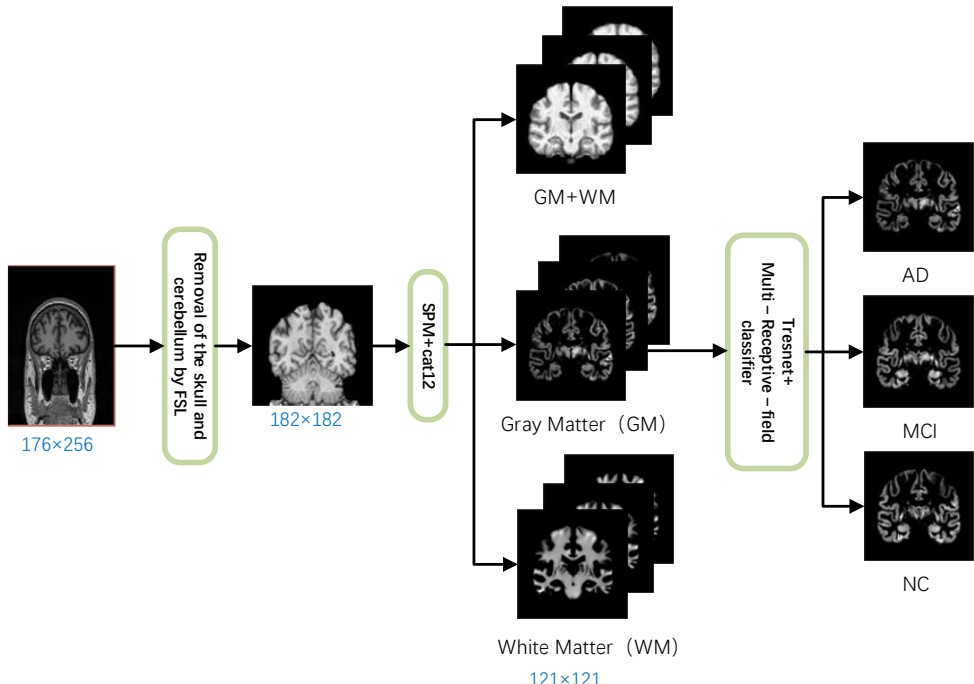

**Figure 1.** AD recognition flowchart of multi-receptive-field classifier based on Gray Matter images.

### 3.1. Data Acquisition

In this article, data acquisition comes from the Alzheimer's Disease Neuroimaging Initiative database (ADNI, adni.loni.usc.edu). The ADNI was founded in 2003 as a public-

private partnership [5,26]. To ensure the independence of data in this study, the public dataset is selected including the Alzheimer's Disease Neuroimaging Initiative (Screening 1.5T), which was collected only once per patient. A total of 85 AD, 244 NC and 133 MCI subjects are selected in this paper. The demographics of the subjects are detailed in Table 1.

**Table 1.** Demographic information of the subjects in this study from ADNI database.

| Daignosis | Age | Sex (Male (M)/Female (F)) |
|-----------|-----|---------------------------|
| AD | $73.63 \pm 7.68$ | 43 M/42 F |
| MCI | $73.82 \pm 7.47$ | 156 M/88 F |
| NC | $75.29 \pm 4.85$ | 75 M/58 F |

*3.2. Data Preprocessing*

3.2.1. MRI Data Processing Flow

The structural MRI images are acquired from 1.5T scanners. We download data in the Neuroimaging Informatics Technology Initiative (NIFTI) forma. Screening 1.5T is used in the structural MRI data provided by ADNI. In this dataset we found that some MRI images of the patients are taken twice when processing the data, and two similar subjects are generated, so the second subject is removed to prevent data leakage in this article. Next the batch command line of FSL software is utilized to skull removal, cerebellum removal, and the registration of heterogeneous images. Then SPM tool kits in MATLAB are utilized for anterior commissure (AC) correction. The Gray Matter (GM), White Matter (WM), and GM+WM are split by cat12 tool kits in the standard space. Compared with WM and CSF, GM has a greater difference between NC and AD [27]. Therefore, the GM image in coronal view is taken as the input of the network model, three slices are taken for each subject. We obtain the MNI152_T1_1mm through average of 152 human brain T1 scan images. The MNI152_T1_1mm is used for the standardized spatial brain template, the obtained slice size is $121 \times 121$. And the length and height are widened to $128 \times 128$. Figure 2 shows GM, WM, and GM+WM in axial, coronal, and sagittal views, respectively.

3.2.2. The Choice of the Most Informative Slices

In the selection of slices, each subject is extracted slices of the same sequence in coronal view after standardization for ensuring that the features of each category of slices are the same and effective. At the same time, in order to increase the number of deep learning dataset, each subject is extracted three adjacent slices. According to the human brain shown in [3], the slices of hippocampus and its surrounding structures can be well displayed, as shown in Figure 2b,e,h. Slices that can well show the hippocampus and surrounding structure. There are 20 consecutive slices of a subject that can well show the hippocampus and the surrounding structure in coronal view, and we choose the middle 3 slices from 20 consecutive slices. Figure 3 shows the sample MRI of Alzheimer's disease (AD), mild cognitive impairment (MCI), and Normal Control (NC) in coronal view from (a) to (c), respectively.

*3.3. Methodology*

The convolutional neural network is used to simulate the activity of the human cerebral cortex, and to convert the low-level input tensor features into high-level features through convolutional layers one after another. In general, a deep CNN architecture consists of an input layer, a number of hidden layers followed by an output layer. The input form is a tensor, and the tensor size equals that of the input 2D image data multiplied by batchsize. The high-level features are obtained through several convolutional layers and the pooling layers, then through the fully connected layer; the distribution probability of each image in each type is obtained in the output layer. The input tensor includes dimension of image as well as the number of images for one training session. For example, the dimension of the input tensor of 2D CNN is three, and the dimension of the input

tensor of 3D CNN is four. In Figure 1, it shows a schematic diagram of the classification process in this article. It takes 2D image in coronal view as input, since three slices are extracted from each subject, and the dataset has a total of 462 subjects, that means a total of 1386 images are obtained. Taking into account the size of the dataset and the computer configuration, the batchsize is set to 16 in order to facilitate the training of the model.

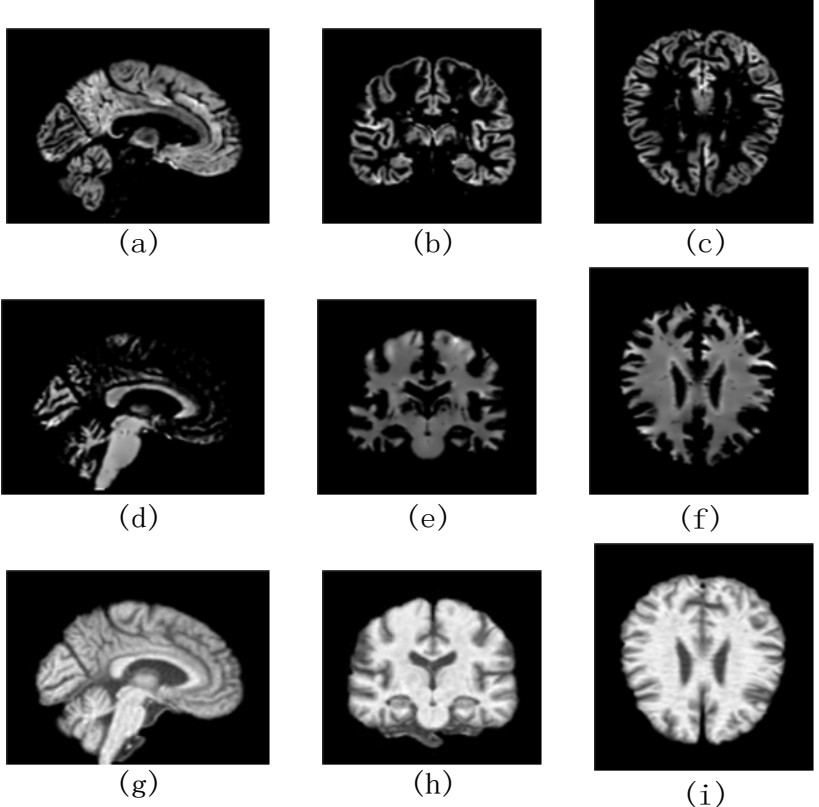

**Figure 2.** Slices of an AD patient, from **left** to **right**: in axial view, coronal view, and sagittal view, from **up** to **down**: GM, WM, and GM + WM, respectively.

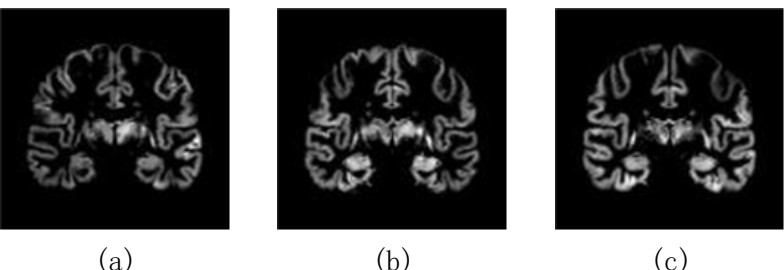

**Figure 3.** The sample MRI of AD, MCI, and NC in coronal view from (**a**–**c**).

3.3.1. Tresnet

An improved network based on the Tresnet network model is proposed. The Tresnet network was proposed by the DAMO Academy (Alibaba Group) in March 2020 [28]. The network had higher accuracy and efficiency than the previous ConvNets. According to the depth and the number of channels, There are three variants of Tresnet network: Tresnet_M, Tresnet_L, and Tresnet_XL.

Tresnet retains the traditional BasicBlock and Bottleneck modules in the basis of Resnet, and is modified in the following five aspects: SpaceToDepth stem, Anti-Alias downsampling (AA), In-Place Activated BatchNorm, Blocks selection, and SE module.

SpaceToDepth: SpaceToDepth stem is added to reduce the amount of calculation through decreasing the resolution, which can greatly decrease the calculation time of the training model. The stem of SpaceToDepth is shown in Figure 4:

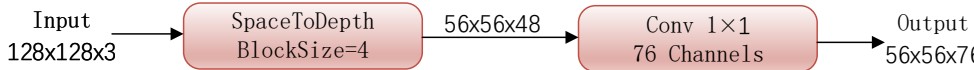

**Figure 4.** Tresnet stem design.

Anti-Alias downsampling (AA): AA is used to replace all downsampling layers. All stride_2 convolutions in the downsampling layer are replaced by stride_1 convolutions, which are followed by a stride_2 $3 \times 3$ fixed blur filter. The equivalent AA component is used to replace all downsampling layers in the network to improve the translational equidistance of the deep network. AA is shown in Figure 5:

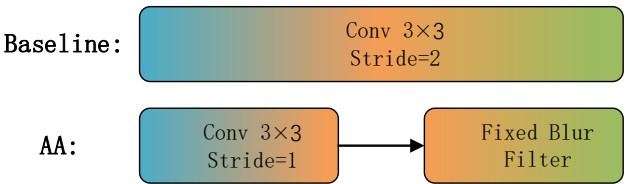

**Figure 5.** Anti-Alias downsampling(AA).

In_Place Activated BatchNorm (Inplace_ABN): The BatchNorm + ReLU layer is replaced by the Inplace_ABN layer in the entire Tresnet network. This layer is responsible for fusing BatchNorm and activation as a single operation, which greatly reduces the memory required for training the deep network. At the same time, the little increase in the amount of calculation can be ignored in cost. As the activation function of Inplace_ABN, Leaky_ReLU is used instead of the ordinary ReLU activation function. Using Inplace_ABN instead of BatchNorm can double the maximum processing quantity, and using Leaky_ReLU instead of ReLU activation function can make the model achieve better accuracy under the same amount of calculation.

Block Selection: Generally speaking, Bottleneck has a higher GPU usage than BasicBlock, but it has a higher accuracy. Mixing the BasicBlock layer and the Bottleneck layer can provide better balance of speed and accuracy. BasicBlock layer has a larger receptive field, it is usually more effective at the beginning of the network. Therefore, the BasicBlock layer is placed on stage 1 and stage 2 of the network, and the Bottleneck layer is placed on stage 3 and stage 4.

SE module: A dedicated SE module is added to the Tresnet architecture. The SE module mainly includes Squeeze, Excitation and Scale submodules. And the SE module is placed in the first three stages of the network. Compared with the standard SE module, the placement and hyperparameters of the SE have also been optimized in the Tresnet. And the reduction factor of the SE module is set to 8 in the Bottleneck unit; the reduction factor of the SE module is set to four in the BasicBlock unit. The structure of SE module is shown as in Figure 6. In Figure 6, X is the 3D tensor of input, C1 and C2 represent the number of channels.

### 3.3.2. SK Module

In a standard convolutional network, the size of the receptive fields of neurons in each layer is fixed. In neurology, the size of the visual neuron's receptive fields is constructed by the stimulus mechanism. However, this factor is rarely considered in convolutional networks. The Selective Kernel (SK) module can make the neuron adaptively adjust the size of its receptive field for input information of different sizes. The SK module combines SE operator, Merge-and-Run Mappings, and Attention mechanism on inception block ideas.

The specific implementation of SK module is divided into three steps: Split, Fusion and Selection. Through these three operations, the characteristics of different receptive fields are obtained, and the weighted characteristics are fused [29]. Figure 7 is the SK module flow chart:

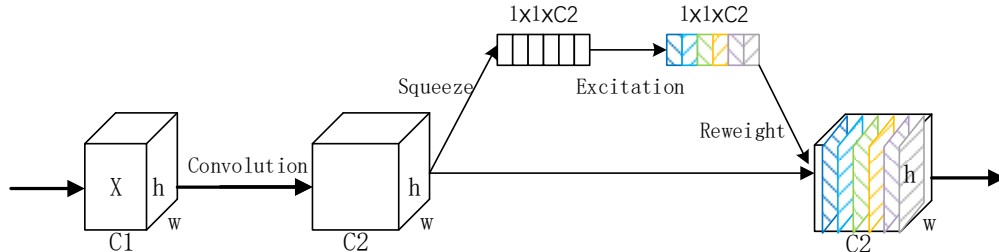

**Figure 6.** SE module.

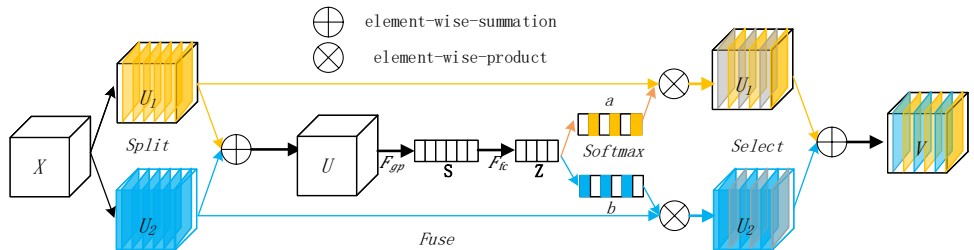

**Figure 7.** Selective Kernel Convolution.

Split: Split is a series of convolution operations with different sizes of convolution kernels on the vector X to obtain several different output matrices. As shown in the structure diagram, the input X is convolved with the convolution kernel size of $3 \times 3, 5 \times 5$, respectively, and the output matrices $U_1$ and $U_2$ are obtained.

Fusion: This part is a combination of different characteristics obtained from different receptive fields, $U = U_1 + U_2$. $F_{gp}$ is a global average pooling operation, and $F_{fc}$ denotes two fully connected layers where the first reduces dimensionality and the next increases dimensionality. It should be noted that the two output matrices $a$ and $b$, where Matrix $b$ is a redundant matrix, in the case of two branches as shown in the Figure 7, satisfy $b = 1 - a$.

Selection: The selection operation corresponds to the scale in the SE module. The difference is that in selection stage two weight matrices, $a$ and $b$, are used to perform weighting operations on $U_1$ and $U_2$, finally the weighted $U_1$ and $U_2$ are summed to get the output vector $V$.

$$V = aU_1 + bU_2 \tag{1}$$

Compared with the SE, the SK module has multiple branches from input to output, and the soft attention mechanism is used to integrate the information from different receptive fields. Different receptive fields make the generalization ability of the entire network better. At the same time, although the number of branches increases, the SK module uses fewer theoretical parameters and floating-point operations per second (FLOPS) by group convolution, so the number of parameters between the SK and SE modules are almost equal in this paper. Actually, the number of parameters of Trsenet_L reaches 52.3 MB, and the number of parameters of Tresnet_L+SK reaches 52.0 MB. The fusion of the receptive fields from different branches makes more generalized features learned. In the medical field, due to the limitation of collecting MRI data, the amount of collected data is small. Therefore, the slight increase in the amount of calculation is not important in training and verifying the neural network.

In this paper, the Tresnet network is used as a basic network, and each SE module in the first three stages is replaced by a SK module. Better training and testing accuracies of

the preprocessed MRI images for 462 subjects could be achieved than those in the basic Tresnet. Among them, Tresnet_L+SK has achieved the best accuracy. Table 2 shows the structure of the Tresnet_L+SK network:

**Table 2.** The network structure of Tresnet_L+SK module.

| Layer | Block | Output | Stride | Tresenet+SK | |
|---|---|---|---|---|---|
| | | | | **Repeats** | **Channnels** |
| Stem | SpaceToDepth | $56 \times 56$ | - | 1 | 48 |
| | Conv $1 \times 1$ | | 1 | 1 | 76 |
| Stage 1 | BasicBlock+SK | $56 \times 56$ | 1 | 4 | 76 |
| Stage 2 | BasicBlock+SK | $28 \times 28$ | 2 | 5 | 152 |
| Stage 3 | Bottleneck+SK | $14 \times 14$ | 2 | 18 | 1216 |
| Stage 4 | Bottleneck | $7 \times 7$ | 2 | 3 | 2432 |
| Pooling | GlobalAvgPool | $1 \times 1$ | 1 | 1 | 2432 |
| Params | | | | 52.0 M | |

### 3.3.3. Transfer Learning

Supervised learning requires a large amount of already labeled data. Labeling data is a boring and costly task, and in reality, there is not a large number of data to be trained, so transfer learning is adopted more and more [30].

Transfer learning means to transfer the trained model knowledge to the new model to help the new model training. The knowledge that has been learned can be shared with the new model through transfer learning to speed up and optimize the learning efficiency of the model.

In our research, considering that the MRI dataset is small and the data distribution is imbalanced, we first use a part of the ImageNet public dataset to pretrain the proposed model, and then transfer the obtained model parameters to train the MRI dataset of this article.

## 4. Evaluation Metrics and Experimental Results

Researches in recent years have shown that the effect of AD diagnosis by deep learning is usually better than the traditional method of manually designing features under the same conditions of data [30,31]. We test the proposed deep learning framework based on the screening dataset without data leakage, which includes 462 ADNI subjects (85 AD, 244 MCI and 133 NC subjects). As we all know that overfitting may occur when we train a model by small dataset, we randomly divide the entire MRI data into five groups in order to prevent overfitting. The distribution of AD, MCI, and NC in any group is consistent with the percentage distribution of AD, MCI, and NC in the entire MRI data. Use one group as the test set, and the remaining four groups as the training set for five-fold cross-validation.

When using small and imbalanced dataset, in order to evaluate the classification ability of the proposed model, the average of five-fold cross-validation experiments is used as the final experimental result, and the evaluation indexes are accuracy, sensitivity (SEN), and specificity (SPE). The accuracy is the ratio of correctly classified samples to the total number of samples, and it can be calculated by the Formula (2). Formulas (3)–(5) are the calculation methods of sensitivity, specificity, and F1-score (F1), respectively.

$$Accuracy = \frac{TP + TN}{TP + TN + FP + FN} \tag{2}$$

$$Sensitivity = \frac{TP}{TP + FN} \tag{3}$$

$$Specificity = \frac{TN}{TN + FP} \tag{4}$$

$$F1 = \frac{2 * Sensitivity * Accuracy}{Sensitivity + Accuracy} \tag{5}$$

$TP$: True Positive, the prediction result is positive and the actual result is also positive.

$FP$: False Positive, the prediction result is positive but the actual result is negative.

$FN$: False Negative, the prediction result is negative but the actual result is positive.

$TN$: True Negative, the prediction result is negative and the actual result is also negative.

In this paper, we consider binary classification and three-way classification of tasks, namely AD subjects versus NC subjects (AD vs. NC) and AD subjects versus MCI subjects versus NC subjects (AD vs. MCI vs. NC), respectively. In the mean time, the 5-fold cross-validation method is used to compare the traditional CNN (LeNet) with the newer networks in recent years, such as Resnet, Mobilenetv2, Densenet, Tresnet, and so on. The effects of these models we test are shown in the Tables 3 and 4.

**Table 3.** The accuracy of recent classification networks on AD vs. NC and AD vs. MCI vs. NC tasks.

| Network Model | AD vs. NC | AD vs. MCI vs. NC |
|---|---|---|
| LeNet | 79.5% | 55.6% |
| Resnet | 82.1% | 58.6% |
| Mobilenetv2 | 82.6% | 59.1% |
| Densenet | 83.9% | 58.8% |
| Tresnet | 84.8% | 58.2% |
| Tresnet+SK | 85.9% | 61.8% |

**Table 4.** The SEN, SPE and F1 of recent classification networks on AD vs. NC tasks.

| Network Model | SEN | SPE | F1 |
|---|---|---|---|
| LeNet | 76.5% | 82.6% | 78% |
| Resnet | 79.2% | 85.9% | 80.6% |
| Mobilenetv2 | 77.2% | 89.1% | 80% |
| Densenet | 81% | 88.6% | 82.4% |
| Tresnet | 81.9% | 88% | 83.3% |
| Tresnet+SK | 82.1% | 88.3% | 83.9% |

On the basis of the Tresnet network which has the best effect, SK module is added into the network to integrate the characteristics of different receptive fields and improve the classification effect. The effects of several variants of Tresnet and Tresnet+SK we test are shown in the Tables 5 and 6.

Meanwhile, in order to speed up and optimize the training of the model, the transfer learning method based on ImageNet dataset is adopted. Transfer learning speeds up the convergence of the model, and enhances the performance of the model.

**Table 5.** Accuracy of several variants of Tresnet.

| Model | AD vs. NC | AD vs. MCI vs. NC |
|---|---|---|
| Tresnet_M | 84.4% | 58.2% |
| Tresnet_M+SK | 84.8% | 59.7% |
| Tresnet_L | 84.9% | 58.9% |
| Tresnet_L+SK | 85.9% | 61.8% |
| Tresnet_XL | 84.1% | 57.6% |
| Tresnet_XL+SK | 84.8% | 58.3% |

**Table 6.** The SEN, SPE and F1 of several variant of Tresnet.

| Model | SEN | SPE | F1 |
|---|---|---|---|
| Tresnet_M | 80.4% | 87.5% | 82.4% |
| Tresnet_M+SK | 81.5% | 87.5% | 83.1% |
| Tresnet_L | 80.9% | 87.3% | 82.9% |
| Tresnet_L+SK | 82.1% | 88.3% | 83.9% |
| Tresnet_XL | 80.4% | 87.1% | 82.2% |
| Tresnet_XL+SK | 80.6% | 87.7% | 82.7% |

## 5. Discussion

According to the above experimental results, our proposed method can not only automatically segment the GM from MRI, but also intercept the images including the hippocampus and surrounding structures in the coronal view. Based on the use of the latest classification network Tresnet, the characteristics of different receptive fields are integrated, and the classification accuracy is further improved under the same depth of neural network. On the ADNI dataset without data leakage, the proposed GM multi-receptive-field classification method achieves an accuracy of 86.9% in AD vs. NC, and 63.2% in AD vs. MCI vs. NC. As shown in Tables 7 and 8, this method is better than the traditional 2D CNN network in diagnosing AD tasks, and also better than many 3D CNN methods. Some methods based on SVM get better results than our method. However, in the case of big dataset, the manual feature extraction of SVM method has a large workload, and SVM is a binary classifier, its multi classification is also composed of binary classification. So SVM is difficult application in big dataset. The method we propose is based on deep learning, and it has the potential to deal with the big dataset.

**Table 7.** Performance comparison of the proposed method with the previous methods.

| Method | Dataset | AD vs. NC | 3-Ways |
|---|---|---|---|
| Korolev et al. [20] 3D CNN | 50 AD + 120 MCI + 61 NC | 80.0% | - |
| Valliani et al. [21] 2D CNN | 188 AD + 243 MCI + 229 NC | 81.3% | 56.8% |
| Cheng et al. (MRI) [22] 3D CNN | 193 AD + MCI + NC | 85.5% | - |
| Zhang et al. [16] SVM | 38 AD + 42 MCI + 40 NC | 87.7% | 84% |
| Wen et al. [25] 2D CNN 3D subject-level CNN 3D path-level CNN SVM | 336 AD + 787 MCI + 330 NC | 82% 83.5% 78% 88% | - |
| Lin et al. [23] 3D multi-model | 193 AD + 151 NC | 77% | - |
| Kanghan et al. [24] 3D CNN | 198 AD + 230 NC | 86.6% | - |
| Proposed | 85 AD + 244 MCI + 133 NC | 86.9% | 63.2% |

As for the classification effect of the network model, Tables 3 and 4 show that our proposed network model is better than many popular deep learning models, such as LeNet, Resnet, Mobilenetv2, Densenet, and so on. In order to verify the effect of GM image compared with WM and GM+WM image, we test the effect of three types of image

classification, respectively. Tables 9 and 10 show these classification effects, GM is better than WM and GM+WM. The proposed classification method is tested on the ADNI dataset consisting of 85 AD, 244 MCI and 133 NC subjects. The classification accuracies of AD vs. NC and AD vs. MCI vs. NC are 85.9% and 61.8%, respectively, thus showing the robustness of this method.

**Table 8.** SEN, SPE and F1 comparison of the proposed method with the previous methods.

| Method | Dataset | SEN | SPE | F1 |
| --- | --- | --- | --- | --- |
| Korolev et al. [20] | 50 AD + 120 MCI + 61 NC | 79.3% | 73.9% | 79.6% |
| Cheng et al. (MRI) [21] | 193 AD + MCI + NC | 83.8% | 90% | 84.6% |
| Kanghan et al. [24] | 198 AD + 230 NC | 88.5% | 84.5% | 92.8% |
| Proposed | 85 AD + 244 MCI + 133 NC | 84% | 88.7% | 85.4% |

**Table 9.** The accuracy of Tresnet integrating multi-receptive-field mechanism in GM, WM, and GM+WM.

| Model | Tresnet+SK | |
| --- | --- | --- |
| | AD vs. NC | AD vs. MCI vs. NC |
| GM+WM | 82.9% | 56.2% |
| WM | 82.1% | 56.9% |
| GM | 85.9% | 61.8% |

**Table 10.** The SEN and SPE of Tresnet integrating multi-receptive-field mechanism in GM, WM, and GM+WM.

| Model | Tresnet+SK | | |
| --- | --- | --- | --- |
| | SEN | SPE | F1 |
| GM+WM | 78.9% | 82.1% | 80.9% |
| WM | 78.3% | 83.1% | 80.2% |
| GM | 82.1% | 88.3% | 84.4% |

Compared with most other datasets of today, the number of medical images is less, especially there are much fewer datasets in AD due to the difficulty of collection. In order to increase the amount of data and maintain the effect of the data, we collect consecutive slices for each subject. On the other hand, the accuracy and speed of training after GM segmentation are different due to the differences in GM and WM. In theory, the smaller the dataset, the faster the training. Figure 8 shows the result of Tresnet_L with pretraining and without pretraining, where the pretraining parameters are trained on the ImageNet dataset. As shown in Figure 8, the model is set for a total of 60 epochs without pretraining and with pretraining, respectively. At the 25th epoch, the accuracy reaches the highest without pretraining. In the case of pretraining, the training accuracy is higher totally, and reaches the highest accuracy more quickly, and it starts to stabilize basically at the eighth epoch. Figure 8 shows that transfer learning accelerates the convergence of training and optimizes the training effect.

As shown in Figure 9, even if the proposed model does not use pretraining parameters, but it obtains better performance than the Tresnet networks with pretraining model. Although the training speed of our proposed Tresnet+SK model is not as fast as the above-mentioned three types of Tresnet networks with pretraining parameters, but its accuracy is the highest, which reaches 85.9%. Furthermore, transfer learning is added for comparison in the proposed model. The model parameters trained on the ImageNet dataset are transferred to the MRI dataset. As shown in Table 11, the effect of the trained model after transfer learning is better.

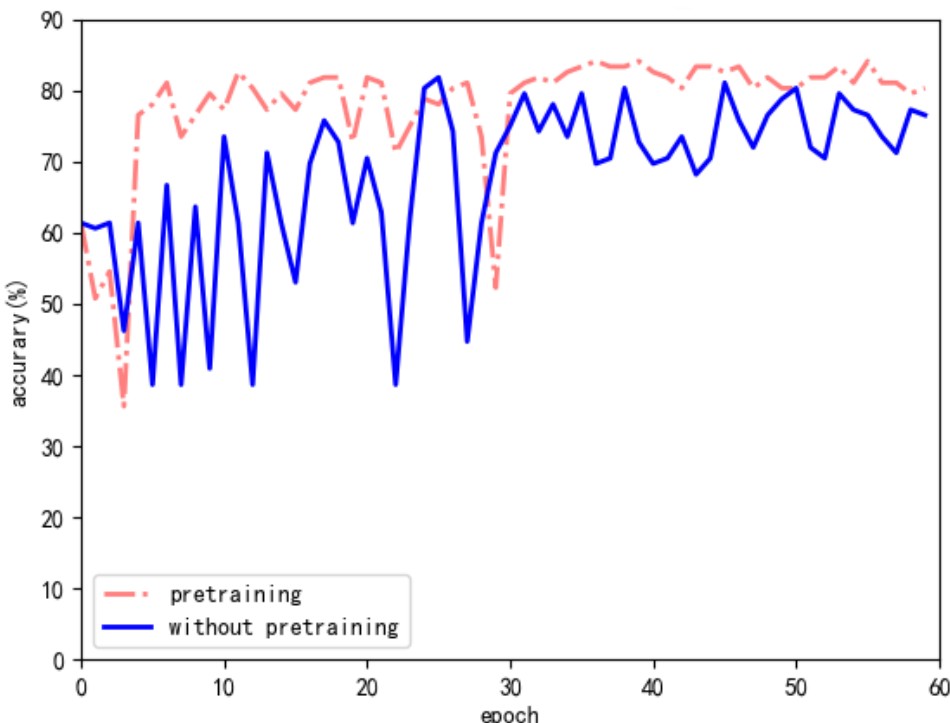

**Figure 8.** Accuracy of the proposed model without pretraining and pretraining Tresnet_L.

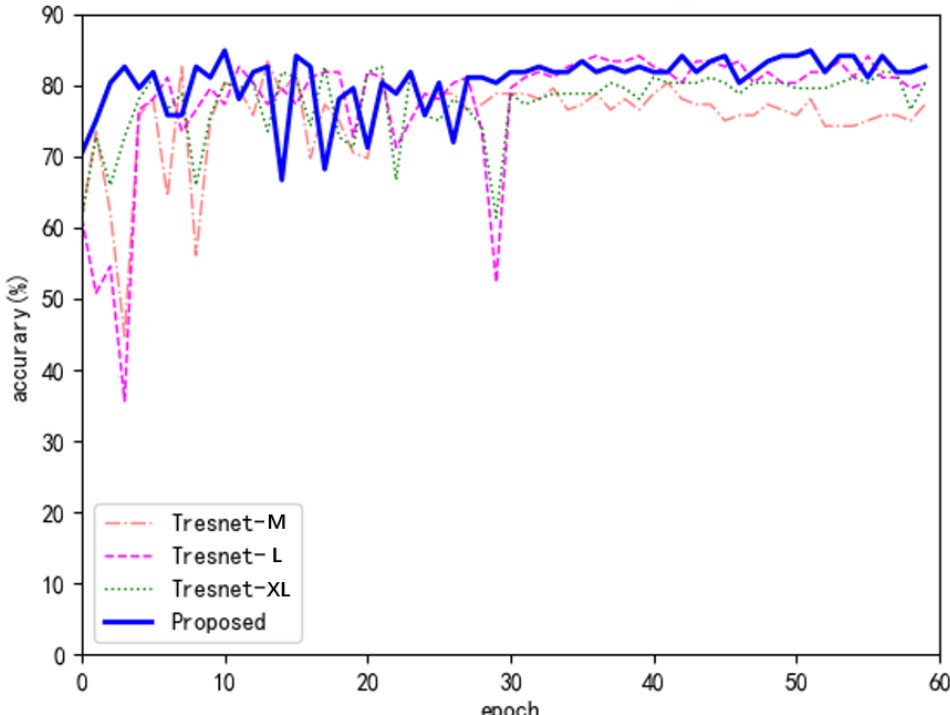

**Figure 9.** Accuracy of three types of Tresnets with pretraining and the proposed model without pretraining.

Figure 10 shows the result of Tresnet_L+SK pretraining on the ImageNet dataset and migrating the trained parameters to the MRI images in this article.

**Table 11.** Classification effect of Tresnet+SK.

| Model | AD vs. NC | AD vs. MCI vs. NC |
|---|---|---|
| without pretraining | ACC: 85.9%<br>SEN: 82.1%<br>SPE: 88.3%<br>F1: 83.9% | ACC: 61.8% |
| pretraining | ACC: 86.9%<br>SEN: 84%<br>SPE: 88.7%<br>F1: 85.4% | ACC: 63.2% |

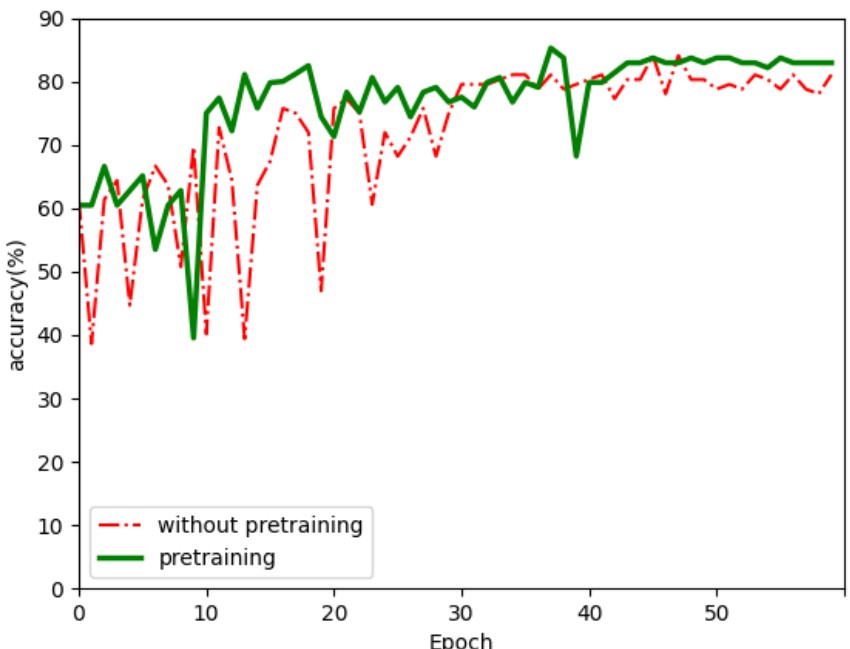

**Figure 10.** Accuracy of Tresnet_L+SK with pretraining and without pretraining.

## 6. Conclusions

We propose a novel framework based on the modified Tresnet neural network to diagnose AD, which uses the most informative images from the coronal view to distinguish among three types of situations: AD, MCI, and NC. Firstly, based on the MRI data from ADNI website, the Gray Matter images with the largest differences among the three types namely WM, GM, and GM+WM for training, validating and testing the neural network model. Secondly, the MRI data of this paper are trained in several recent classification models and compared with the previous researches. The Tresnet is used as the basic model in this article based on our results of performance comparison among several convolutional deep neural networks. Thirdly, the SK module is used to replace the SE module in the first three stages of Tresnet integrating the characteristics of different receptive fields to improve the classification effect. Finally, due to the small and imbalanced dataset, transfer learning method is used to train the model for speeding up and optimizing the learning efficiency.

For diagnosing AD, to our best knowledge, the training results of our model are better than those of the existing 2D CNN and many 3D CNN model frameworks without data leakage, which shows the effectiveness of diagnosing AD. In our future work, we plan to applied the proposed method on more datasets and training models based on extracting region of interest (ROI) from MRI dataset. We will further devoted to the actuate prediction from MCI to AD, which is valuable for early predicting and preventing the conversion to AD.

**Author Contributions:** Conceptualization, methodology, software and writing—original draft preparation, Z.X.; editing and supervision, H.D.; visualization Y.Y. and J.L. All authors have read and agreed to the published version of the manuscript.

**Funding:** This work is partially supported by National Natural Science Foundation of China (Grant No.62020106010).

**Conflicts of Interest:** The authors declare no conflict of interest.

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
