# Peer review of "Diagnosis of Alzheimer’s Disease Based on the Modified Tresnet"

_electronics, doi:10.3390/electronics10161908_

Round 1
Reviewer 1 Report
In the manuscript a deep learning-based methodology is proposed for Alzheimer’s Disease diagnosis and discrimination among AD and another dementia type. The methodology is based on brain images from a public available dataset. The objective is clear and of scientific interest. Overall, the manuscript is well structured and well written in general; however; several sentences need revisions and I strongly suggest a careful proof reading from a native English speaker. Below are some comments:
- Diagnosis of Alzheimer’s Disease is not based particularly on MRI rather than on clinical neurological and psychological tests and is confirmed through EEG and MRI. Thus, I suggest the authors to include in the first paragraph of the 2nd page this statement and refer to the recent studies below:
- a) Tzimourta, K. D., Christou, V., Tzallas, A. T., Giannakeas, N., Astrakas, L. G., Angelidis, P., ... & Tsipouras, M. G. (2021). Machine Learning Algorithms and Statistical Approaches for Alzheimer's Disease Analysis Based on Resting-State EEG Recordings: A Systematic Review. International Journal of Neural Systems, 2130002-2130002.
- b) Yang, S., Bornot, J. M. S., Wong-Lin, K., & Prasad, G. (2019). M/EEG-based bio-markers to predict the MCI and Alzheimer's disease: a review from the ML perspective. IEEE Transactions on Biomedical Engineering, 66(10), 2924-2935.
- Line 139. The appropriate reference for the ADNI database is missing.
- The authors should clearly state the novelty of the proposed methodology.
Author Response
Dear reviewer:
Thank you for your valuable comments. We have thought about these comments thoroughly, and revised our manuscript carefully.
Best regards,
Yours sincerely,
Zelin Xu,
On behalf of all the authors

Reviewer 2 Report
The topic seems interesting, I have the following concerns to enhance the quality of the work.
- Your title is too long and should be reduced. A good title should not exceed 10 words.
The title should be clear and informative and should reflect the aim and approach of the work.
Recommendations for titles:
- Fewest possible words describe the contents of the paper.
- Avoid waste words like "Studies on", or "Investigations on”, “effects of”, “comparison of”, or “a case of”
- Use specific terms rather than general
- Watch your word order and syntax
- Avoid abbreviations and jargon
- Authors should revise the abstract and accuracy should be added at end of the abstract.
- Why tensor network gives the best performance?
- I am not satisfied with the introduction of the paper. The introduction should provide a clearer view of the paper, as paper belong to computer-aided diagnoses so authors should think about it and should add few CAD papers in introduction and related work section
Moradi, E., Pepe, A., Gaser, C., Huttunen, H., Tohka, J. and Alzheimer's Disease Neuroimaging Initiative, 2015. Machine learning framework for early MRI-based Alzheimer's conversion prediction in MCI subjects. Neuroimage, 104, pp.398-412. Khan, M. A., & Kim, Y. (2021). Cardiac arrhythmia disease classification using LSTM deep learning approach. CMC-COMPUTERS MATERIALS & CONTINUA, 67(1), 427-443. Li, S., Shi, F., Pu, F., Li, X., Jiang, T., Xie, S. and Wang, Y., 2007. Hippocampal shape analysis of Alzheimer's disease based on machine learning methods. American Journal of Neuroradiology, 28(7), pp.1339-1345.
- The research problem/ requirement is not elaborated properly.
- Fig1 should need to redraw.
- The Authors need to explain how to handle class imbalance. It must be added to the proposed method.
- Authors need to re-write the Abstract in a more meaningful way example (Problem definition=> How existing methods are lacking => proposed solution => Outcome
- All equations should be assigned numbers. And align with the text.
- All figures should be redrawn with high resolutions and different colors.
Conclusion and Future work must be updated.
Author Response

(The authors gave the same response as above.)

Reviewer 3 Report
Dear authors,
First of all, I would like to congratulate to the authors for this work. Absolutely, it is an important topic in the medical field using a deep learning approach as a strategy to find solutions on Alzheimer’s disease. I have considered that it is a great advance in this area, a novel framework and that it will have a great improvement. This paper is perfectly written, understandable, and well organized. I have learnt a lot with your proposal, and the literature review is excellent. The proposed solution is really well explained and a high quality of scientific impact.
The only suggestion is if the authors will open access the code and dataset in a GitHub repository or similar. I think that is important to share as an open source this work. For this reason, it will be the only requirement that is missing in this work. Authors can add a little section with the link. Indeed, it is an important aspect also to evaluate this work. I am really impressed of this work and I like this version to be accepted for publish with this requirement.
Best regards,
Author Response

(The authors gave the same response as above.)

Round 2
Reviewer 1 Report
The authors answered all the comments and performed all the necessary changes in the manuscript.